# Case Report—An Inherited Loss-of-Function *NRXN3* Variant Potentially Causes a Neurodevelopmental Disorder with Autism Consistent with Previously Described 14q24.3-31.1 Deletions

**DOI:** 10.3390/genes14061217

**Published:** 2023-06-02

**Authors:** René G. Feichtinger, Martin Preisel, Karin Brugger, Saskia B. Wortmann, Johannes A. Mayr

**Affiliations:** 1University Children’s Hospital, Salzburger Landeskliniken (SALK) und Paracelsus Medical University (PMU), 5020 Salzburg, Austria; r.feichtinger@salk.at (R.G.F.); m.preisel@salk.at (M.P.); kar.brugger@salk.at (K.B.); h.mayr@salk.at (J.A.M.); 2Amalia Children’s Hospital, Radboudumc, 6525 HB Nijmegen, The Netherlands

**Keywords:** *NRXN3*, neurodevelopmental disorder, autism, novel disease

## Abstract

Background: Heterozygous, large-scale deletions at 14q24.3-31.1 affecting the neurexin-3 gene have been associated with neurodevelopmental disorders such as autism. Both “de novo” occurrences and inheritance from a healthy parent suggest incomplete penetrance and expressivity, especially in autism spectrum disorder. *NRXN3* encodes neurexin-3, a neuronal cell surface protein involved in cell recognition and adhesion, as well as mediating intracellular signaling. *NRXN3* is expressed in two distinct isoforms (alpha and beta) generated by alternative promoters and splicing. MM/Results: Using exome sequencing, we identified a monoallelic frameshift variant c.159_160del (p.Gln54AlafsTer50) in the *NRXN3* beta isoform (NM_001272020.2) in a 5-year-old girl with developmental delay, autism spectrum disorder, and behavioral issues. This variant was inherited from her mother, who did not have any medical complaints. Discussion: This is the first detailed report of a loss-of-function variant in *NRXN3* causing an identical phenotype, as reported for heterozygous large-scale deletions in the same genomic region, thereby confirming *NRXN3* as a novel gene for neurodevelopmental disorders with autism.

## 1. Introduction

Neurexins were discovered as the major alpha-latrotoxin receptor and vertebrate-specific toxin of the black widow spider (*Latrodectus*) [1]. The human neurexin family consists of three unrelated genes (*NRXN1*, *NRXN2*, and *NRXN3*) [2]. Neurexins are highly expressed in presynaptic nerve terminals and have important roles in synaptic cell adhesion and neurotransmitter release [3]. 

Interestingly, in contrast to *NRXN1* and *NRXN3*, *NRXN2* unexpectedly represses excitatory synapse assembly in the hippocampus, in which neurexins can be divided into pro- and antisynaptogenic organizers [4]. A number of *NRXN3* isoforms, alpha-type and beta-type, are produced by the use of alternative promoters and alternative splicing (https://www.uniprot.org/uniprotkb/Q9HDB5/entry, https://gtexportal.org/home/gene/NRXN3; accessed on 1 April 2023) [5,6,7,8]. The larger, α isoform contains additional exons on the 5’ site [9]. The more abundant β isoforms contain no EGF-like repeats (epidermal growth factor like repeats), but the six laminin/neurexin/sex hormone-binding globulin (LNS) domains, which are found in α isoforms [10]. Additionally, the *NRXN1* gene can encode an even shorter neurexin γ isoform [8,11]. Extensive alternative splicing is a hallmark of the NRXN family. Six alternative splicing sites in the α isoforms, and two (shared with α form) in the β isoforms give rise to more than a thousand isoforms, explaining the extreme phenotypic variance in patients and mouse models [7,12,13]. Interestingly, autoantibodies against NRXN1α and NRXN3α might also play a role in the development of encephalitis, schizophrenia-related phenotypes, amnestic cognitive impairment, and depressive symptoms [14,15,16,17,18].

Rare exonic deletions within *NRXN1* (MIM #600565) at 2p16.3 are among the most frequently observed copy number variations (CNVs) in autism spectrum disorder (ASD) [2,19,20,21,22,23,24,25,26]. The spectrum of heterozygous deletions in *NRXN1* includes ASD, ADHD (attention deficit hyperactivity disorder), intellectual disability, seizures, schizophrenia, mood disorders, and congenital malformations [27]. Variants in *NRXN2*, although less frequent, have been associated with autism and language delay [28,29]. Moreover, the deletion of *Nrxn2* in mice results in autism-related behaviors [30,31]. Other members of the neurexin family are contactin-associated protein-like 1 (*CNTNAP1*), also termed neurexin 4, and contactin-associated protein-like 2 (*CNTNAP2*). Variants in *CNTNAP1* can cause hypomyelinating neuropathy, congenital, 3 (MIM# 618186), and lethal congenital contracture syndrome 7 (MIM# 616286) [32]. For both disorders, no autistic phenotype was reported, despite delayed development and absent speech being described. *CNTNAP2* is required for the radial and longitudinal organization of myelinated axons (https://www.uniprot.org/uniprotkb/Q9UHC6/entry). Variants in CNTNAP2 are reported to cause Pitt–Hopkins-like syndrome 1 (MIM#610042). This autosomal recessive syndrome is characterized by developmental delay, speech delay, and seizures, as well as autism [33,34]. In addition, *CNTNAP*3-5, which belongs to the neurexin family, was reported to be associated with autism [32]. Neurexin interaction partners such as neurolig-in 1 (NLGN1) (MIM#618830), neuroligin 3 (NLGN3) (MIM#300425), and the calcium/calmodulin-dependent serine protein kinase (CASK) (MIM#300749, 300422) are reported in neurodevelopmental disorders and/or autism etiology and susceptibility [35,36]. Heterozygous deletions of chromosomal region 14q24.3-31.1 involving *NRXN3* have been associated with various neuropsychiatric conditions (e.g., ADHD or schizophrenia) and autistic features with/without developmental delay/intellectual disability in a total of 24 individuals from 14 families [37,38,39,40]. No other NDD-linked genes were frequently deleted in 14q24.3.31.1 microdeletion syndrome patients. In 6/14 (43%) of the families, the deletion was dominantly inherited from one parent; in 6/14 (43%) it was *de novo*, while in 2/14 (14%), the mode of inheritance was not available. Four families showed deletions of exons of the β-isoform; two were inherited from their fathers, and two were *de novo* [2,41]. The remarkable plethora of different isoforms also partially explains the phenotypic diversity. 

Several knock-out mouse models have been generated to analyze the function of the NRXN family. Nrxn1/Nrxn2/Nrxn3 triple knock-out mice revealed that at least two intact α-Nrxn genes are necessary for survival [3,42]. Here, brain morphology was normal at birth; however, the mice showed an impairment of excitatory and inhibitory synaptic transmission. Several constitutional and conditional mouse models revealed a role of β-Nrxn3 in neurotransmitter release, excitatory postsynaptic current, induction of action potential, AMPAR (α-amino-3-hydroxy-5-methyl-4-isoxazolepropionic acid receptor) levels, synapse number, and behavior [43,44,45,46]. It was shown that in Nrxn3 conditional knock-out mice, alternative splicing at SS2 and SS4 regulates the release probability but not the number of inhibitory synapses in the olfactory bulb [47]. Exon 24 encodes an evolutionarily conserved glycosylphosphatidylinositol (GPI) anchor site [48]. Importantly, despite evidence from affected patients with copy number variations, *NRXN3* is, to date, not listed as a monogenic disease-relevant gene in OMIM (https://omim.org/entry/600567). Recently, according to ClinVar, a total of 120 *NRXN2* variants have been identified, and among those, 5% have ASD [29,49,50,51].

## 2. Material and Methods

### 2.1. Whole-Exome Sequencing

We performed exome sequencing from leucocyte-derived DNA from the affected child (singleton exome). The library was prepared by SureSelect60Mbv6 (Agilent) and paired-end sequenced on a HiSeq 4000 platform (Illumina), with a read length of 100 bases [52]. In order to align reads to the human genome assembly hg19, a Burrows-Wheeler Aligner (BWA, v.0.5.87.5) was applied and detection of genetic variation was performed using SAMtools (v 0.1.18), PINDEL (v 0.2.4t), and ExomeDepth (v 1.0.0). The cut-off for biallelic inheritance was set to < 1% allele frequency, while for monoallelic inheritance, it was set to < 0.1%. The number of reference entries was 26,174 exomes in the database at the time of analysis. Samples achieved a 20× coverage of 95.5%. Reads were aligned to the human genome-build GRCh37/hg19 and assessed for sequence alterations using a custom-made bioinformatics tool [52]. Sanger sequencing was performed using standard methods. No array CGH was performed (Figure 1A).

### 2.2. Web Resources/Tools and Databases Used for the Current Study

DECIPHER; https://www.deciphergenomics.org [53]

ClinVar; https://www.ncbi.nlm.nih.gov/clinvar [54]

gnomAD; https://gnomad.broadinstitute.org [55]

OMIM; https://www.omim.org

UCSC Genome Browser; https://genome.ucsc.edu [56]

UniProt; https://www.uniprot.org [57]

GTEx Portal; https://gtexportal.org

Varsome; https://varsome.com [58]

Denovo-db; https://denovo-db.gs.washington.edu

Genematcher; https://genematcher.org [59]

## 3. Results

### 3.1. Clinical Description

This girl is the second child of healthy, non-consanguineous Pakistani parents. Her older sister and younger brother have no medical complaints. Her pregnancy, delivery, anthropometric data at birth, and postnatal adaptation were unremarkable. 

She achieved independent walking at the age of 20 months but cannot run or climb at her current age of 5^0/12^ years. She is unresponsive when called by name and does not follow any commands. She does not make any syllables or words nor does she use gestures. She only communicates discomfort by crying. An assessment of her hearing with audiometry was impossible due to non-compliance, but brainstem response evoked audiometry (BERA) showed normal results. She does not make eye contact or seek physical contact with her parents or other individuals. 

Her play consists of emptying drawers. She is not toilet-trained and wears diapers. The physical and neurological examinations were unremarkable. The only somatic complaint was constipation, responding well to macrogols. Based on these observations the Austrian developmental screening tool from Jentschura und Janz and the Autism Diagnostic Observation Schedule (ADOS), the diagnosis of a global developmental delay and autism spectrum disorder was made [60].

### 3.2. Genetic Testing

An autosomal recessive search (frequency < 500 cases) in our in-house exome database containing 26,174 individuals gave 161 non-synonymous variants. An autosomal dominant search (frequency < 5 cases) revealed 302 non-synonymous variants. A total of 38 variants (dominant search) affected genes with a pLI score of 1. Furthermore, 28 CNVs (250 controls allowed) were detected. 

An exome analysis revealed a monoallelic frameshift variant in *NRXN3* NM_001272020.2 (c.[159_160del];[=] (p.[Gln54AlafsTer50];[=]), which was absent from gnomAD (Figure 1B). Additionally, no further cases with this variant were present in the in-house cohort. The *NRXN3* variant was confirmed by Sanger sequencing and found to be inherited from the mother. According to the ACMG (American College of Medical Genetics) guidelines, the variant is classified as likely pathogenic (PVS1, PM2, PP4, and BS2). Alternative promotor usage produces several α and β isoforms [61]. The protein encoded by the β promotor contains a signal sequence of 34 amino acids. The promotor for the β isoforms is located downstream of exon 17. The frameshift mutation affects the 15th amino acid after a signal sequence and, therefore, should lead to a lack of all β isoforms.

**Figure 1 genes-14-01217-f001:**
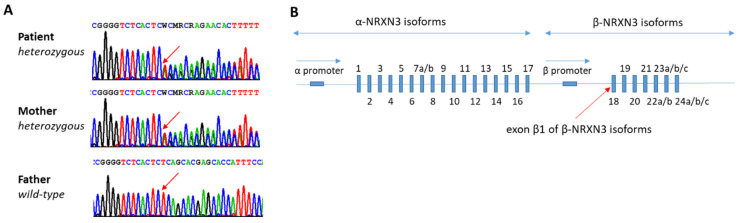
Results of Sanger sequencing and schematic presentation of *NRXN3* gene organization. (**A**): Sanger sequencing of the affected patient and the parents. (**B**): Schematic presentation of the *NRXN3* gene organization showing the two promoters used for generation of long α- and short β-NRXN3 isoforms adapted from Hishimoto et al. (2007) [61]. The *NRXN3* LoF variant affects exon β1. The p.Gln54AlafsTer50 variant should cause a complete loss of all *NRXN3* β-isoforms in the patient.

## 4. Discussion

Numerous microdeletion disorders were published prior to the identification of the specific disease genes residing within the critical deletion region. Initially, 9q deletions were linked to Kleefstra syndrome (9q Subtelomeric Deletion Syndrome, 9q34.3 Microdeletion Syndrome, 9qSTDS), and later, it was recognized that single nucleotide variants (SNVs) in the euchromatic histone methyltransferase 1 (*EHMT1*) gene cause Kleefstra syndrome (MIM#607001) [13]. Further, notable examples include Van-Asperen syndrome/chromosome 17q11.2 deletion syndrome (MIM#613675)l harboring *NF1* (Neurofibromatosis 1) (MIM#613113); Koolen–De Vries syndrome, harboring *KANSL1* (KAT8 regulatory NSL complex, subunit 1) (MIM#610443); Miller–Dieker syndrome (MIM#247200), harboring *PAFAH1B1* (platelet-activating factor acetylhydrolase, isoform 1B, alpha subunit) (MIM#601545); and Smith–Magenis Syndrome/17p11.2 microdeletion syndrome, including *RAI1* (retinoic acid-induced gene 1) (MIM#182290) [62,63].

Here, we provide further evidence that *NRXN3* is likely the responsible disease gene associated with the phenotype observed in chromosome 14q24.3-31.1 deletions in both human and knock-out mouse models [2,10,41,46]. Reported symptoms include intellectual disability, ADHD, motor delay, aggression, depression, anxiety, facial dysmorphism, language delay, social impairment, learning difficulties, schizophrenia, temper tantrums, color blindness, self-harm obsessions, sleep issues, obesity, delusional and persecutory ideas, seizures, and hypotonia. The individual presented here is the first reported with a maternally inherited SNV in *NRXN3* that is associated with an NDD with behavioral features. In gnomAD, six loss-of-function (LoF) variants have been reported. In the “gnomAD v2.1.1. (controls)” subset (60,146 out of 141,456), three LoF variants remain. One variant lies within a non-coding transcript, and the other two affect the last exon, potentially allowing the production of stable transcripts. In summary, there might be no LoF variants in gnomAD controls, pointing to the extremely rare nature of *NRXN3* LoF variants. Furthermore, in gnomAD, the following LoF variants are listed: c.1833_1849+5dupAAATCTGGATTTGAAAGGTAAA, c.2497+2_2497+3dupTA p.Arg639Ter, and p.Arg1013GlyfsTer26. The variant p.Arg1013GlyfsTer26 affects the last exon of *NRXN3* and might give rise to a stable transcript. For another variant, c.1833_1849+5dupAAATCTGGATTTGAAAGGTAAA, it is unclear whether it influences splicing. In total, there are two potential LoF candidate variants: p.Arg639Ter and c.2497+2_2497+3dupTA. In the database at https://denovo-db.gs.washington.edu, no loss-of-function variants are reported. In DECIPHER, a maternally inherited and likely pathogenic LoF variant (p.Arg176Ter) has been reported [53] in a patient with autistic behavior, delayed speech and language development, and an intellectual disability. We contacted the responsible physician to obtain some additional information. For the individuals mother a speech delay and a mild intellectual disability was reported. No autism was present, once more underlining that penetrance plays an important role in the development of autism in NRXN3 deficiency. A very short clinical description has been published recently [64]. The clinical features of the individual reported in DECIPHER fit well with those of our patient.

Recently, a patient with a *NRXN3* missense variant of uncertain significance (p.Arg39Cys) and a neurodevelopmental disorder was described [65]. Additionally, a microdeletion of 7177 base pairs (79176037–79183213) affecting *NRXN3* was detected in four patients; two of them were sisters in a Saudi epilepsy cohort [66]. Furthermore, two Iranian families were described with homozygous and compound heterozygous variants characterized by learning disabilities, developmental delays, an inability to walk, and behavioral problems [67]. 

The variant identified in our study was also deposited on Genematcher. However, no proper matches were present on the Genematcher platform [59]. In detail, totally different clinical features and/or low ACMG ratings, low conservation of the affected residues, and another mode of inheritance of the Genematcher entries led to exclusion. No matches with sufficient evidence were identified.

The clinical features of the two patients (the one presented here and the one in [64]). are similar to those reported in individuals with 14q24.3-31.1 deletions. Although all three variants affect highly conserved residues, prediction algorithms classify the two missense variants, p.Arg1332His (rs200707419) and p.Arg1481Gln (rs768341004) (NM_001330195.2, NP_001317124.1), as “variants of unknown significance” based on PM2 and PP2. The individual carrying the p.Arg1481Gln was compound heterozygous for c.3142 + 3A > G (rs531047390). This splice site mutation scores well in accordance with the ACMG criteria. Analyses of additional patients with an autosomal recessive inheritance are required to demonstrate that *NRXN3* can both be inherited by autosomal recessive and the described autosomal dominant mode as seen in the chromosome 14q24.3-31.1 deletions. Additionally, six patients with biallelic *NRXN1* mutations were identified, raising the question of whether all neurexin-related disorders (NRXN1, NRXN2, and NRXN3) can be inherited via an autosomal recessive or autosomal dominant mode [34,68,69]. 

A total of 24 individuals with heterozygous deletions affecting either part or the entire *NRXN3* gene have been reported [2,10,41,46]. In six families, the deletions were not detectable in parental leucocyte-derived DNA, and presumably occurred *de novo*. In another six families, these were inherited from a (healthy) parent, suggesting incomplete penetrance and expressivity, especially for autism [2,41]. Further, the mother in our study did not display any overt symptoms. However, a detailed and complete neuropsychological evaluation was not performed. Therefore, it cannot be excluded that some subtle abnormalities are present. In Sanger sequencing, no signs of a mosaic were obvious, which could explain the lack of symptoms in the mother. No exome sequencing was performed for the mother.

Consistently, it is reported that for the paralogue *NRXN1*, reduced penetrance and variable expressivity remain challenges for genetic counselling. The estimation of penetrance in previous studies for *NRXN1* was between 10.4 and 62.4% for different CNVs. For a large cohort including 67 individuals from 34 families (11 paternal inheritance; 10 maternal inheritance), cascade screening of 71 first-degree relatives of the 21 inherited cases led to the identification of an additional 24 carriers, of whom 13 showed the typical phenotype. Speech delay is the most consistent clinical finding in individuals harboring the NRXN1 deletion. It has been estimated that the penetrance of several key features manifests as global delay (83%), speech delay (94%), intellectual disability (80%), autism spectrum disorder (60%), and seizures (17%) [27]. NRXN1 deletions can be inherited from apparently healthy parents [53]. In addition, incomplete penetrance was observed for variants in *CNTNAP2* with presentation in heterozygous carriers ranging from severe to apparently unaffected [70].

In summary, all disorders caused by variants in members of the neurexin family seem to show an incomplete penetrance and/or expressivity that underscore our results. In two families, the mode of inheritance was not available. The main phenotype of our individual (autism spectrum disorder and speech delay) is in line with previously reported cases and mouse models. A fundamental similarity to the clinical features observed in *NRXN1* and *NRXN2* patients is also present. 

## 5. Conclusions

In summary, heterozygous CNVs encompassing *NRXN3*, intragenic deletions, and specific monoallelic SNVs in *NRXN3* are associated with a neurodevelopmental disorder with behavioral/autistic features. We therefore propose that *NRXN3* is a potential novel causative disease gene for neurodevelopmental disorder/autism spectrum disorder. 

## Data Availability

The data presented in this study are available on request from the corresponding author. The data are not publicly available due to institutional guidelines.

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
