# Peer review of "Case Report—An Inherited Loss-of-Function NRXN3 Variant Potentially Causes a Neurodevelopmental Disorder with Autism Consistent with Previously Described 14q24.3-31.1 Deletions"

_genes, 2023, doi:10.3390/genes14061217_

Round 1
Reviewer 1 Report
The authors identified a dominant frameshift variant affecting the beta isoform of NRXN3 in a patient reported to have developmental delay and ASD. According to them, it "confirms" that NRXN3 pathogenic variants cause "an identical phenotype" (line 21) to the 14q24-31 deletion phenotype. I do not think that such a strong statement can be made given the information provided in the manuscrit for the following reasons :
1- this is a single case
2- many other variants were identified following exome sequencing in this patient
3- analysis and familial segregation of the other variants is not presented
4- given the numerous transcripts existing for the beta isoform of NRXN3, a confirmation that no beta mRNA remains would be convincing
5- 14q deletions may cause color blindness, facial dysmorphism or obesity, all absent from the reported patient. Hence the case does not explain all the 14q34-31 deletion phenotype.
My understanding is that the patient does not speak or use volontary movements at 5 years of age (Lines 123 to 126). Please convince the reader that the ASD diagnosis scales you use are adapted to the current case, otherwise revise the classification of this phenotype.
I do not understand how the exome sequencing results were analyzed. What is an in-house database of 5 control cases (line 139) ? Please provide numbers according the population databases. Also discuss the possible role of the other variants identified in this patient. A better evaluation of their possible involvement (including segregation analysis) could have been made to convince the readers further that NRNX3 is the only explanation to the phenotype of this patient.
Given the numerous transcripts, can you exclude that a beta-isoform transcript persists in the patient despite the SNV identified ? Is it possible to analyze mRNAs ?
The statement that NRXN3 explains all of the 14q24-31 deletion phenotype is too strong. The authors state that 14q34-31 deletions may cause color blindness, facial dysmorphism or obesity, all absent from their patient. For me this means that NRXN3 does not explain it all.
The discussion on the presence of LoF alleles in interesting but misleading. The authors disregard their possible presence in control individuals and I do not understand why since they have themselves identified a control individual with a LoF allele (the mother of their patient). This section of the discussion must be modified (Lines 172 to 176).
The paper needs english editing.
Small details
Line 29 : please rephrase "unlinked genes" do you mean "unrelated" ?
Line 73: a variant is "de novo" not "inherited de novo"
Line 91: I do not understand the sentence "However, concerning the entry in OMIM 91 the same applies to NRXN2. "
Figure 1: you can provide the IGV screenshot or the Sanger trace. Both elements are not needed (Sanger preferred). Panel C is not necessary given the simple familial structure.

The paper needs english editing.
Author Response
The authors identified a dominant frameshift variant affecting the beta isoform of NRXN3 in a patient reported to have developmental delay and ASD. According to them, it "confirms" that NRXN3 pathogenic variants cause "an identical phenotype" (line 21) to the 14q24-31 deletion phenotype. I do not think that such a strong statement can be made given the information provided in the manuscrit for the following reasons :
1- this is a single case
We are aware of the limitations of a single case. We tried to temper the message a bit. However, we think that it is important to publish case reports to pave the way for the diagnosis of patients that will be identified in future.
2- many other variants were identified following exome sequencing in this patient
All the other variants were excluded for numerous reasons. Please find some of the more important below.
- clinical features do not fit to the disease (isolated cardiomyopathy, isolated cataract, syndromes with severe dysmorphic features etc can be excluded)
- furthermore variants affecting positions with a low phylogenetic conservation were excluded
- furthermore variants that are likely benign or benign according to the ACMG classification were excluded
In summary, only the NRXN3 variant remained as the diagnosis in the patient.
3- analysis and familial segregation of the other variants is not presented
We briefly explained in point 2 that no other variants were identified that could explain the phenotype and have a good ACMG rating.
4- given the numerous transcripts existing for the beta isoform of NRXN3, a confirmation that no beta mRNA remains would be convincing
In principle we agree that these experiments would be easy to perform. However, NRXN3 is almost exclusively expressed in the brain. This was also a reason why it is a good candidate for isolated autism It is not expressed in blood. Please see: https://gtexportal.org/home/gene/NRXN3
5- 14q deletions may cause color blindness, facial dysmorphism or obesity, all absent from the reported patient. Hence the case does not explain all the 14q34-31 deletion phenotype.
This is true, but that additional symptoms are present in patients with large 14q deletions encompassing numerous additional genes is clear. However, the smallest region of overlap was determined for 14q deletions as NRXN3.
My understanding is that the patient does not speak or use volontary movements at 5 years of age (Lines 123 to 126). Please convince the reader that the ASD diagnosis scales you use are adapted to the current case, otherwise revise the classification of this phenotype.
This is indeed a patient that does not speak, but that she cannot make voluntary movements is not correct. As we describe she can walk and play etc. She is not able to communicate with words or gestures. The test batteries used are appropriate to test children as the one described and testing was performed by experienced neuropsychologists. We have added the citation for the ADOS for more clarity.
Lord C, Risi S, Lambrecht L, Cook EH Jr, Leventhal BL, DiLavore PC, Pickles A, Rutter M. The autism diagnostic observation schedule-generic: a standard measure of social and communication deficits associated with the spectrum of autism. J Autism Dev Disord. 2000 Jun;30(3):205-23. PMID: 11055457.
I do not understand how the exome sequencing results were analyzed. What is an in-house database of 5 control cases (line 139) ?
In the Center for Genomics in Munich whole exome sequencing was performed for 27000 individuals. The data of all of those are available in the in house data base. This database is used for analysis of the exome results, but also other databases as gnomAD and Kaviar are implemented.
We show the analysis for autosomal dominant inheritance. In the given example we said, show all variants that are found in 5 samples of the in house individuals or less. In this case 302 variants remain for analysis. However, we also had a look at more frequent variants but this would be too much information and confuse the reader.
Please provide numbers according the population databases.
The NRXN3 variant is not present in gnomAD and the in house data base. The information is given in line 147.We also included the following sentence was included in line 146: “Also no further case with this variant was present in the in house cohort“
Also discuss the possible role of the other variants identified in this patient. A better evaluation of their possible involvement (including segregation analysis) could have been made to convince the readers further that NRNX3 is the only explanation to the phenotype of this patient.
As stated for point 2 there are no other variants explaining the phenotype.
Given the numerous transcripts, can you exclude that a beta-isoform transcript persists in the patient despite the SNV identified ? Is it possible to analyze mRNAs ?
As stated before NRXN3 is predominantly expressed in brain.
The statement that NRXN3 explains all of the 14q24-31 deletion phenotype is too strong. The authors state that 14q34-31 deletions may cause color blindness, facial dysmorphism or obesity, all absent from their patient. For me this means that NRXN3 does not explain it all.
We tried to temper the message a bit. This is true, but that additional symptoms are present in patients with 14q deletions encompassing numerous additional genes is clear. However, the smallest region of overlap was determined for 14q deletions as NRXN3. We tried to set the focus on the autism spectrum disorder that is the key feature of the disorder.
The discussion on the presence of LoF alleles in interesting but misleading. The authors disregard their possible presence in control individuals and I do not understand why since they have themselves identified a control individual with a LoF allele (the mother of their patient). This section of the discussion must be modified (Lines 172 to 176).
There might be some carriers in gnomAD but no affected individuals. If these individuals will get children with autism is totally unclear and speculative.
The paper needs english editing.
An English editing was done by a native speaking colleague.
Small details
Line 29 : please rephrase "unlinked genes" do you mean "unrelated" ?
We changed the wording according to the reviewers suggestion.
Line 73: a variant is "de novo" not "inherited de novo"
The wording was corrected.
Line 91: I do not understand the sentence "However, concerning the entry in OMIM the same applies to NRXN2. "
We rephrased the sentence: Also NRRXN2 is not listed in OMIM as a disease gene.
Figure 1: you can provide the IGV screenshot or the Sanger trace. Both elements are not needed (Sanger preferred). Panel C is not necessary given the simple familial structure.
According to reviewer 1 we deleted panel A and C. According to reviewer 2, we included red arrows to point out were the variant occured.
Feichtinger and fellow authors present a case report of a 5 year old individual with a novel NRXN3 variant. Since previous reports have mainly examined large genomic deletions, this genetic variant in NRXN3 suggests a potential important role for this gene in the symptoms described in the larger gene deletion patients. Overall, this is a unique case report that will further the impact of NRXN3 dysfunction in humans. I only have some minor suggestions for the authors:
1) line 54. Please change "results in autism spectrum disorder" to "results in autism related behaviors"
We changed the sentence according the reviewer.
2) line 70. use ADHD here since it was defined earlier in line 51
We used ADHD according to the reviewer.
3) Should there be a description of how the BERA (brainstem response evoked audiometry) in the methods section?
The responsible physician answered: This is a standardized hearing test that does not need further description of the method in the context of this article, especially as it was unremarkable.
4) line 164. While the authors report suggests NRXN3 is likely the gene causing symptoms in 14q24.3-31.1 syndrome, the "the responsible disease gene" text should be tempered by replacing with "a likely responsible disease gene"
We tempered the text by using likely.
5) For figure 1. Can the figure be rearranged to make panel A bigger? Also, it would help to have some annotation in panel B to point out where the variant occurs in the patient and mother.
According to reviewer 1 we deleted panel A and C. According to reviewer 2, we included red arrows to point out were the variant occured.
Reviewer 2 Report
Feichtinger and fellow authors present a case report of a 5 year old individual with a novel NRXN3 variant. Since previous reports have mainly examined large genomic deletions, this genetic variant in NRXN3 suggests a potential important role for this gene in the symptoms described in the larger gene deletion patients. Overall, this is a unique case report that will further the impact of NRXN3 dysfunction in humans. I only have some minor suggestions for the authors:
1) line 54. Please change "results in autism spectrum disorder" to "results in autism related behaviors"
2) line 70. use ADHD here since it was defined earlier in line 51
3) Should there be a description of how the BERA (brainstem response evoked audiometry) in the methods section?
4) line 164. While the authors report suggests NRXN3 is likely the gene causing symptoms in 14q24.3-31.1 syndrome, the "the responsible disease gene" text should be tempered by replacing with "a likely responsible disease gene"
5) For figure 1. Can the figure be rearranged to make panel A bigger? Also, it would help to have some annotation in panel B to point out where the variant occurs in the patient and mother.
Author Response
Feichtinger and fellow authors present a case report of a 5 year old individual with a novel NRXN3 variant. Since previous reports have mainly examined large genomic deletions, this genetic variant in NRXN3 suggests a potential important role for this gene in the symptoms described in the larger gene deletion patients. Overall, this is a unique case report that will further the impact of NRXN3 dysfunction in humans. I only have some minor suggestions for the authors:
1) line 54. Please change "results in autism spectrum disorder" to "results in autism related behaviors"
We changed the sentence according the reviewer.
2) line 70. use ADHD here since it was defined earlier in line 51
We used ADHD according to the reviewer.
3) Should there be a description of how the BERA (brainstem response evoked audiometry) in the methods section?
The responsible physician answered: This is a standardized hearing test that does not need further description of the method in the context of this article, especially as it was unremarkable.
4) line 164. While the authors report suggests NRXN3 is likely the gene causing symptoms in 14q24.3-31.1 syndrome, the "the responsible disease gene" text should be tempered by replacing with "a likely responsible disease gene"
We tempered the text by using likely.
5) For figure 1. Can the figure be rearranged to make panel A bigger? Also, it would help to have some annotation in panel B to point out where the variant occurs in the patient and mother.
According to reviewer 1 we deleted panel A and C. According to reviewer 2, we included red arrows to point out were the variant occured.
Round 2
Reviewer 1 Report
I have no further comments. I would have liked to see more details concerning the exome sequencing results (other than NRXN3) but unfortunately this is not provided. Saying "there is nothing else" is ok I believe, but providing the actual data would have been a plus (302 variants and 28 CNVs were identified in this patient).
Author Response
I have no further comments. I would have liked to see more details concerning the exome sequencing results (other than NRXN3) but unfortunately this is not provided. Saying "there is nothing else" is ok I believe, but providing the actual data would have been a plus (302 variants and 28 CNVs were identified in this patient).
Reply 1: All variants obtained from exome sequencing are checked at least by 2 independent scientists. Actually, we do not have the permission to publish or provide all genetic variants for a single individual. Especially, since incidental findings not relevant for the disease should not be made availbale to the family. It is an Ethics standard at least in our facility not to communicate about currently irrelevant variants that maybe have an effect later in life. E.g if a child with NDD is administered, we would not communicate a variant in BRCA1. Neither to the child nor to the mother.